# Learning to Discern: Imitating Heterogeneous Human Demonstrations with Preference and Representation Learning

**Sachit Kuhar,**\* **Shuo Cheng, Shivang Chopra, Matthew Bronars, Danfei Xu**
Georgia Institute of Technology
{kuhar, shuocheng, shivangchopra11, mbronars, danfei}@gatech.edu

**Abstract:** Practical Imitation Learning (IL) systems rely on large human demonstration datasets for successful policy learning. However, challenges lie in maintaining the quality of collected data and addressing the suboptimal nature of some demonstrations, which can compromise the overall dataset quality and hence the learning outcome. Furthermore, the intrinsic heterogeneity in human behavior can produce equally successful but disparate demonstrations, further exacerbating the challenge of discerning demonstration quality. To address these challenges, this paper introduces Learning to Discern (L2D), an offline imitation learning framework for learning from demonstrations with diverse quality and style. Given a small batch of demonstrations with sparse quality labels, we learn a latent representation for temporally embedded trajectory segments. Preference learning in this latent space trains a quality evaluator that generalizes to new demonstrators exhibiting different styles. Empirically, we show that L2D can effectively assess and learn from varying demonstrations, thereby leading to improved policy performance across a range of tasks in both simulations and on a physical robot.

**Keywords:** Imitation Learning, Preference Learning, Manipulation

## 1 Introduction

Imitation Learning (IL) allows robots to learn complex manipulation skills from offline demonstration datasets [1, 2, 3, 4, 5]. However, the quality of the demonstrations used for IL will significantly influence the effectiveness of the learned policies [2, 6, 7]. Practical imitation learning systems must amass demonstrations from a broad spectrum of human demonstrators [8, 9], ranging from novices to experts in a given task [10, 11]. Moreover, even experts may complete a task differently, resulting in disparate demonstrations of equal quality [12, 13, 14]. Therefore, an effective IL algorithm must learn from demonstrations of varying quality, each further diversified by the unique skillset of the individual demonstrators.

However, mainstream IL algorithms often make the simplifying assumption that all demonstrations are uniformly ideal [11]. As a result, policies trained with these algorithms may unknowingly learn from suboptimal or even contradictory supervisions [11, 14, 15]. Recent work attempts to estimate expertise by developing unsupervised algorithms to quantify action distribution and variance [6] or actively soliciting new demonstrations that match with a known expert policy [16]. While these methods are general in principle, two critical limitations remain. First, estimating demonstrator expertise without any external quality label is an inherently ill-posed problem. As noted above, demonstrated behaviors can vary drastically across demonstrators and even across different trials from the same demonstrator. Neither action variance nor similarity to an expert is sufficient to capture expertise in such settings. Second, these works only consider state-level features, while behaviors are better represented in temporal sequences.

---

\*work done while at Georgia Institute of Technology.

7th Conference on Robot Learning (CoRL 2023), Atlanta, USA.

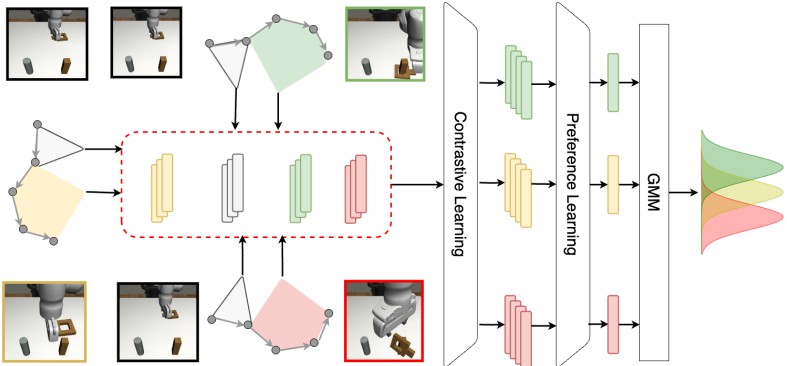

Figure 1: **L2D:** Our framework proceeds in three primary stages during training. First, we augment trajectory segments with temporal embeddings and employ contrastive learning to map these segments to a latent space. Next, we use preference learning in this latent space to train a quality critic on sparse preference labels. Finally, we train a Gaussian Mixture Model (GMM) on the critic's outputs where the different modes represent demonstrator quality.

To this end, we present Learning to Discern (L2D), a completely offline method for learning from heterogeneous demonstrations. L2D can efficiently estimate expertise from limited demonstration quality labels based on sequence-level latent features. By incorporating temporal embeddings into trajectory segments, we are able to learn a more meaningful latent representation for long-horizon manipulation tasks. Preference learning in this latent space trains a quality evaluator that generalizes across diverse styles of task completion while only requiring a subset of coarsely ranked data. We demonstrate that our method can not only discern the quality of in-domain data but also identify high-quality demonstrations from entirely new demonstrators with modes of expertise unseen in training. We show this empirically in both simulated and real-world robot settings. With L2D, it becomes feasible to filter high-quality demonstrations from the vast and diverse datasets that are needed for data-driven IL.

## 2 Related Work

**Imitation Learning with Suboptimal Data.** Imitation Learning (IL) [17, 18, 19, 20, 21, 22] from sub-optimal data is an active area of research in robotics. Various works show that near-optimal policies can be trained if demonstrations are ranked based on their quality [4, 11]. However, manual annotation of dense reward labels is costly and unscalable. CAIL [23] extrapolates demonstration quality from a subset of labeled data $D'$, but still requires a dense ranking of $D'$. Other works estimate demonstrator quality in an unsupervised manner (ILEED [6]) or compare demonstrations to an expert policy (ELICIT [16]). Such unsupervised algorithms do not generalize to equal quality demonstrations from different modes of task completion. This motivates the need for an IL algorithm that learns a generalizable quality estimator from a subset of coarsely ranked data.

**Preference Learning from Human Demonstrations.** Preference learning is a form of supervised learning that learns rankings from pairwise comparisons [24, 25, 26, 27, 28, 29]. When applied to human demonstration data, preference learning can effectively train a reward function based on rankings of trajectory segments [30]. Various frameworks utilize this reward function to train policies through inverse reinforcement learning [31, 32]. To limit manual annotations, some methods synthetically generate lower-quality data through noise injection [7]. These methods are hampered by the fact that noise level does not directly translate to preference ranking of trajectories [33]. Our method, L2D, ranks offline demonstrations using preference learning in a latent space. This allows us to generalize across diverse styles and modes of task completion with sparse preference labels.

**Representation Learning for Data Retrieval.** Representation learning, especially with a focus on robotics, has been widely studied. This area typically explores learning from visual inputs [34], data

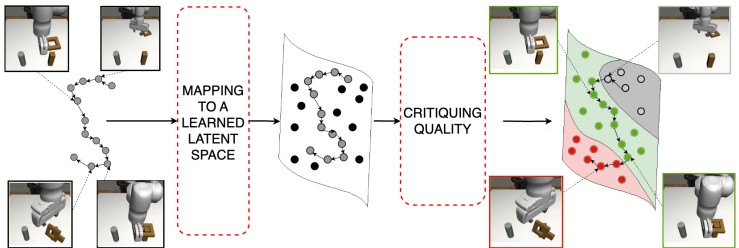

Figure 2: **Filtering Unseen Demonstrations:** When faced with unseen demonstrations, L2D partitions the trajectory into segments and augments each with its chronological ordering in the sequence. The segments are mapped to the latent space learned during training and ranked by the quality critic. After calculating the mean and variance of ranks in a full trajectory, the trained GMM is employed to predict a preference label for the unseen demonstration.

augmentation [35], goal-aware prediction [36], and domain-specific information [37]. Recent work such as that by Du et al. [38] enhances the performance of robot manipulation tasks by first mapping offline data to a latent space using contrastive learning and then retrieving offline data using that latent space with task-specific data as queries. Yet, these works do not account for the suboptimality of given demonstrations in task-specific data and offline data.

## 3 Problem Setting

We study the imitation learning problem in environments modeled as Markov Decision Processes (MDP), where $\mathcal{S}$ and $\mathcal{A}$ denote the state space and the action space, respectively. $P : \mathcal{S} \times \mathcal{A} \times \mathcal{S} \rightarrow [0, 1]$ is the state transition function. The environment emits a binary reward upon task completion. The learning algorithm does not have access to this signal.

We assume access to a small set of demonstration $D_{\text{known}} = \{\tau_i\}_{i=1}^N$, where a trajectory $\tau_i$ is a sequence of transitions with a variable length $L_{\tau_i}$ given by $(s_0, a_0, s_1, a_1, \ldots, s_{L_{\tau_i}-1}, a_{L_{\tau_i}-1})$. Each trajectory in $D_{\text{known}}$ is associated with a quality label $l \in \mathcal{L}$, where $\mathcal{L}$ is the set of possible labels indicating the quality of a trajectory. We divide $D_{\text{known}}$ into sub-datasets i.e., $\{D_A, D_B, D_C, \ldots\}$ of similar quality levels, based on the quality labels associated with each trajectory. If dataset $A$ is considered to be of superior quality compared to $B$, we denote it as $A \succ B$. Similarly, we assume $\tau_A \succ \tau_B$, provided $\tau_A \in A$ and $\tau_B \in B$.

In this paper, we develop a framework for imitation learning that achieves state-of-the-art performance on sub-optimal data by estimating the quality of new demonstrations. More specifically, our aim is to learn a representation that captures the multimodality from a large dataset of unknown quality $D_{\text{unknown}}$ and can critique the quality of new demonstrations within the same context. Importantly, our approach operates in an offline learning setting where we do not have access to the environment, reward signals, or any successful completion of trajectories [6, 1]. This constraint further underscores the need for our method's ability to discern and evaluate the quality of demonstrations without direct interaction or feedback from the environment.

## 4 Learning to Discern

Imitation Learning (IL) with suboptimal data often leads to inferior task performance. Evaluating the quality of a demonstrator without supervision is challenging, and manually assigning quality to demonstrations is impractical. The core component of our method is a preference network $Q$ that learns to evaluate the quality of a demonstration $\tau$ by estimating its ranking label $l$. As mentioned above, the key challenges of learning a generalizable demonstration quality estimator are that (1) it is difficult to capture behavior-level features by assessing individual state-action pairs, especially for long-horizon manipulation tasks and (2) the demonstrated behaviors are diverse even among

demonstrators of similar expertise. In this section, we describe each challenge in more detail and introduce the components in our method L2D that address the corresponding challenges.

**Extracting Behavior Feature through Temporal Contrastive Learning**

As we mentioned earlier, estimating the quality of a demonstration by merely observing variance in states is a poorly-posed problem. Instead, we propose learning a temporal latent space for quality critique. We utilize a neural network encoder $E : \mathbb{R}^{L_1 \times \text{Obs.Dim}} \to \mathbb{R}^{\text{Latent.Dim}}$ that maps trajectory segments $\{\sigma_j = \{s_t\}_{j,t=i}^{t=i+L_1}\}_{j=1}^{i=\text{num\_segments}}$ into a d-dimensional latent space. Sampling triplet pairs for this learning using triplet margin loss [38] poses a challenge: naive sampling based on quality labels may degrade representation quality. Specifically, long-horizon tasks like Square [11] contain bottleneck states indifferent to demonstration quality, leading to task-specific regions that don't majorly impact overall quality. We use these sampled segments to create triplets of an anchor trajectory segment $\sigma_a$, a positive trajectory segment $\sigma_p$, and a negative trajectory segment $\sigma_n$ which are used in different contrastive learning strategies:

- **Strategy 1: Arbitrary Sampling from Specific Quality Sets**: We sample $\sigma_a$ and $\sigma_p$ from different trajectories from a demonstration set $A$. We then sample the negative trajectory segment $\sigma_N$ from a different demonstration quality set $B$.

- **Strategy 2: Specific Segment Sampling from Arbitrary Quality Sets**: We leverage domain-specific knowledge that certain segments, like initial or final segments, do not influence demonstration quality. For instance, in the case of a square task with complete demonstrations, we know that all demonstrations conclude with the robot placing the square in the cylinder. Thus, we sample the final segments $\sigma_{a,final}$ from $\tau^1$ and $\sigma_{p,final}$ from $\tau^2$. The negative trajectory segment $\sigma_n$ is sampled such that it is not from the same region but can be sampled from any subset as the segmented regions themselves do not impact quality.

**Position Encoding for Capturing Non-cyclic Behavior**

Preference learning is effective for evaluating the quality of unknown and diverse demonstrations, but we find it struggles with long-horizon manipulation tasks. We posit that this is due to the intrinsic non-cyclical nature of these tasks can lead to identical state-action pairs having different quality labels depending on the trajectory context. To navigate this inherent complexity, we introduce data augmentations like position encoding for each demonstration state with respect to the entire trajectory. We find that this provides the necessary context to the latent space for comprehensive task understanding. Let our original observation at any time-step $t$ be denoted as $o_t \in \mathbb{R}^{obs.dim}$. We introduce position encoding, represented as a real number $p_t$ indicating the normalized time-step (i.e., $t/T$, where $T$ is the total number of time-steps in the demonstration). We then formulate the augmented observation $o'_t$ which becomes a vector in $\mathbb{R}^{obs.dim+1}$, defined as $o'_t = [o_t, p_t]$.

**Training Quality Critic**

After training the encoder E to learn a latent space that is cognizant of quality-sensitive regions of the demonstrations, our approach leverages a Quality Critic network $Q$, defined as a function $Q : \mathbb{R}^d \to \mathbb{R}$, that serves as a regressor, mapping the d-dimensional trajectory segment embedding to a continuous scalar value indicative of the quality of the demonstration. In the training phase, we utilize pairs of trajectory segments, $\sigma_A$ and $\sigma_B$, randomly sampled from datasets $D_A$ and $D_B$, respectively. Each segment is first converted into an embedding and subsequently passed through $Q$ to obtain a quality score. The pairwise ranking loss [39, 30, 31], capitalizes on these pairs to compare and learn the relative quality differences between the segments. During the inference stage, the model computes the quality score of individual segments.

**Respresenting Suboptimality as a Gaussian Mixture Model**

Demonstrations of varying quality may exhibit mixtures of behaviors in different regions. By being receptive to individual demonstration regions, our approach allows us to estimate the quality of these mixed-quality demonstrations, as the overall quality of a demonstration can be viewed as a majority vote of its constituent parts. This concept can be naturally represented by a Gaussian distribution,

where the mean of the Gaussian represents the average quality of segments in a demonstration. After learning the latent space encoder $E$ and quality critic $Q$, we train a Gaussian Mixture Model (GMM) $G$. We sample a sufficient number of segments ($\lceil \frac{L}{L_1} \times k \rceil$) from each demonstration in $D_{\text{known}}$ and pass each segment through encoder $E$ and quality critic $Q$ to obtain a scalar value. These values represent the quality of different regions within the demonstration. We then map these scalar values into sets that correspond to demonstrations of different qualities (i.e., $D_A$, $D_B$, $D_C$, etc) and train the GMM where different Gaussians correspond to different qualities.

**Estimating Quality of Unseen Demonstrations**

Given a new unseen demonstration $\tau$, we generate a set of scalar values using $Q(E(\text{sampler}(\tau)))$, with each value estimating the quality of a specific region within the demonstration. We then estimate the probability of each value with respect to the trained GMM and assign each scalar to the set with the maximum probability. Finally, we use a heuristic to determine whether the demonstration should be used for training the policy. As an example, in our robomimic experiments, for an unseen demonstration, we count the percentage of segments assigned to the 'good' quality set. Demonstrations are then rated based on this count. We subsequently rank all demonstrations based on this score and select the top demonstrations for training the IL algorithm BC-RNN. This procedure ensures the inclusion of demonstrations that predominantly exhibit desirable behavior.

## 5    Experimental Evaluation

We empirically validate whether L2D can learn effective representations that enable accurate discrimination of high-quality demonstrations from the pool of available demonstrations, which vary in quality. We show that distinct components of the multi-component architecture of L2D significantly contribute to its effectiveness with an ablation study. We demonstrate that L2D can discern high-quality demonstrations from both seen and unseen demonstrators and compare against baselines. Finally, we collect human demonstrations for real and simulated manipulation tasks to test the method's applicability in more realistic scenarios.

**Experiment setup.**  We use the Robomimic [11] for all experiments, which include a range of tasks like Lifting, Square Nut Assembly, and Can Pick-and-Place collected from multiple human operators of varying skill levels. We specifically focus on the Square Nut Assembly task where Robomimic provides us with 300 demonstrations, with 50 from each of the 6 operators. These 300 demonstrations are categorized into 3 sets of 100 demonstrations each and are labeled as 'good', 'ok', or 'bad' based on the execution quality. Using higher quality demonstrations for training leads to a higher success rate in robomimic tasks [11]. To evaluate the different filtering methods, we use RNN-based behavioral cloning (BC-RNN) to train a policy on selected demonstrations and report the success rate (SR) (see appendix D for details).

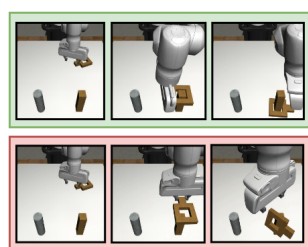

Figure 3: Good (green) and Bad (red) demonstrations for the Robomimic's Square task.

We bifurcate the identification of high-quality demonstrations into two scenarios: *(A) Familiar demonstrators.* The demonstrations in each set originate from the same demonstrators, i.e., both $D_{\text{known}}$ and $D_{\text{unknown}}$ have demonstrations each for varying quality from the same demonstrators. *(B) Unseen demonstrators.* The demonstrations in each set originate from different demonstrators, i.e., both $D_{\text{known}}$ and $D_{\text{unknown}}$ have demonstrations each for varying quality, but they are provided exclusively from different demonstrators. We adapt the robomimic dataset for these scenarios as follows: In both cases, we divide the 300 available demonstrations into two sets of $D_{\text{known}}$ and $D_{\text{unknown}}$. In the first case, we perform this division uniformly such that all 6 operators provide an equal number of demonstrations into either set. However, in the second case, the demonstrations in each set originate from different demonstrators, i.e., both $D_{\text{known}}$ and $D_{\text{unknown}}$ have 50 demonstrations each for varying quality, but they are provided exclusively from different demonstrators.

**Baselines.** We benchmark the performance of our proposed method, L2D, against two state-of-the-art methods for learning from mixed-quality demonstrations. ILEED [6] employs an unsupervised expertise estimation approach to identify good demonstrations. ELICIT [16] actively filters new demonstrations based on whether the state-action pairs match the prediction of an expert policy. We also compare against adaptations of preference learning and contrastive learning methods. Preference Learning baseline is a best effort adaptation of TREX [31] reward learning method and uses pairwise ranking loss [30, 40]. Contrastive Learning uses triplet margin loss samples from distinct quality sets [38] for training and performs filtering by using cosine-similarity between encodings of trajectory segments. As additional baselines, we compare against a naive uniform sampling strategy from the dataset of unknown quality demonstrations, as well as an oracle baseline representing the ideal scenario where the highest-quality demonstrations can be perfectly identified.

## 5.1 Component Ablation

In our ablation study, we dissect L2D to evaluate the individual contribution of each component. By comparing the Wasserstein distances between the distributions corresponding scores generated by $Q$, for sub-dataset with specific labels within $D_{\text{unknown}}$, we quantify the quality of the representation learned. A higher Wasserstein distance indicates enhanced separability between trajectories of different qualities, making it easier to identify high-quality trajectories. In Robomimic tasks, better quality demonstrations result in a higher SR when training a policy using BC methods [11]. Therefore, higher total Wasserstein distances indicate more effective learned representations for filtering.

Table 1 and Figure 4 show our findings and the corresponding distribution visualizations, reveal that L2D consistently outperforms the preference learning approach in unseen demonstrator experiment setup 5.2. The histogram visualizations plot the distributions of trajectory segment scores output by $Q$ for each quality subset in $D_{\text{unknown}}$. Adding the S1 Contrastive component and positional encoding enhances the quality of the learned latent space, confirming the necessity of contextual understanding for long-horizon non-cyclic robot manipulation tasks. The final addition of the S2 Contrastive component underlines the benefits of using well-crafted positive and negative pairs for contrastive learning. These pairs, even from demonstrations of differing quality, lead to better representations, highlighting the importance of understanding that some regions of demonstrations do not influence the quality. This lends support to the idea of leveraging commonalities across all demonstrations to regularize our learning process, thereby avoiding overfitting specific demonstrator biases.

| Method | Good vs. Okay ↑ | Good vs. Bad↑ | Okay vs. Bad↑ | Total Distance↑ |
|---|---|---|---|---|
| Preference | 0.130 | 0.143 | 0.147 | 0.420 |
| + S1 Contrastive | 0.337 | 0.213 | 0.414 | 0.964 |
| + Position Encoding | 0.535 | 0.649 | 0.589 | 1.773 |
| + S2 Contrastive | 0.604 | 0.848 | 0.417 | **1.869** |

Table 1: Wasserstein Distance Comparison for the Robomimic Square Task: The table showcases the capability of different methods, including the incremental introduction of contrastive components, to discern among demonstrations of quality levels (good, okay, bad) from unseen demonstrators.

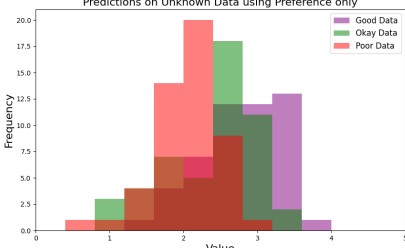 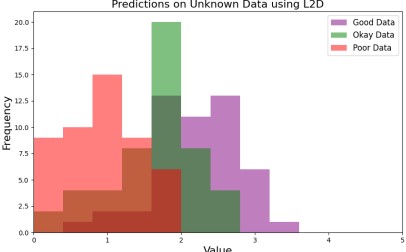

Figure 4: Histogram Distribution Visualization: Comparing demonstration quality scores (good, okay, bad) for the Square task from unseen demonstrators. The left histogram represents a conventional preference learning approach, while the right highlights the efficacy of our method, L2D

## 5.2 Main Results

**L2D can identify unseen high-quality demonstrations from familiar demonstrators.** We trained L2D on the known demonstrations $D_{\text{known}}$ and tested its ability to filter unseen demonstrations $D_{\text{unknown}}$ from the same demonstrators. Our results, as presented in Table 2, indicate that L2D successfully identifies 43 out of the top 50 high-quality demonstrations from the pool. We find that our method outperforms ILEED [6], reinforcing the idea that using preferences over an unsupervised approach leads to better representations. Finally, we observe that our method achieves an oracle-level success rate (SR) when training the policy on filtered data.

| Method | Success Rate |
|---|---|
| Naive | 0.44 |
| ILEED | 0.54 |
| Our Method | 0.66 |
| Oracle | 0.66 |

| Method | Good | Okay | Bad |
|---|---|---|---|
| Naive | 68 | 16 | 16 |
| Our Method | 93 | 4 | 3 |
| Oracle | 100 | 0 | 0 |

Table 2: **Identifying Unseen High-Quality Demonstrations from Familiar Demonstrators:** Comparison of methods to identify high-quality demonstrations in an unseen dataset. The right table presents the quality categorization of selected demonstrations with the best 50 demonstrations from $D_{known}$, while the left displays the imitation learning policy's success rate when trained on a combined set of best-known and selected demonstrations.

**L2D can identify high-quality demonstrations from unseen demonstrators.** In a practical situation, a policy may be trained on data provided by previously unseen demonstrators. This is challenging because each demonstrator may possess a unique style of execution and a diverse range of demonstration quality. We evaluated L2D in this setting by training it and the baselines on $D_{\text{known}}$ and then using them to filter the top 50 demonstrations from $D_{\text{unknown}}$. Table 3 shows that our proposed method L2D significantly outperforms both contrastive learning and preference learning methods, which in turn outperforms naive sampling. In comparison with the ELICIT method [16], which learns only from high-quality data, L2D demonstrates superior performance, suggesting that it is more adaptable to variation in demonstration quality and demonstrator style. Notably, the success rate achieved by our method is only marginally inferior to that of the oracle, further supporting our claim that L2D can effectively discern high-quality demonstrations from unfamiliar demonstrators.

| Method | Success Rate |
|---|---|
| Naive | 0.20 |
| Contrastive | 0.36 |
| Preference | 0.38 |
| Elicit | 0.38 |
| Our Method | **0.44** |
| Oracle | 0.46 |

| Method | Good | Okay | Bad |
|---|---|---|---|
| Naive | 18/50 | 16/50 | 16/50 |
| Contrastive | 23/50 | 20/50 | 7/50 |
| Preference | 27/50 | 20/50 | 3/50 |
| Elicit | 23/50 | 20/50 | 7/50 |
| Our Method | **39**/50 | **8**/50 | **3**/50 |
| Oracle | 50/50 | 0/50 | 0/50 |

Table 3: **Identifying Unseen High-Quality Demonstrations from New Demonstrators:** A comparison of methods to discern top demonstrations from an unfamiliar source. The left table displays the imitation learning policy's success rates, while the right categorizes the quality of demonstrations selected. L2D's performance approaches that of the oracle, highlighting its effectiveness in recognizing quality across unfamiliar demonstrator styles

## 5.3 Learning from Real-World Demonstrations

We evaluate the ability of L2D to estimate the quality of demonstrations in realistic scenarios, compared to controlled simulated environments. We gathered new human demonstrations for tasks such as Square Nut Assembly in simulated environments while Lifting and Stacking were demonstrated in real-world environments using actual robots, all performed by a diverse group of operators. These operators performed independently, without leveraging knowledge from others' demonstrations. We then trained our method on a subset of these demonstrations and used it to filter the unseen dataset

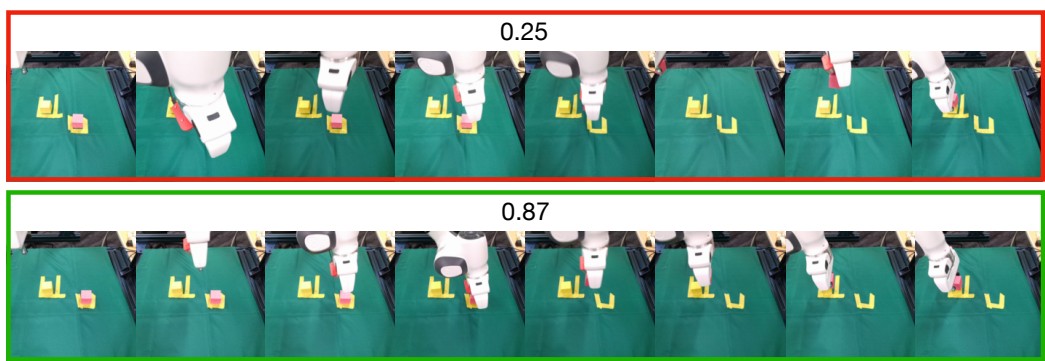

Figure 5: **Demo Quality Estimation.** We show the predicted quality for real-world demonstrations of the Stack task. The trajectory segments with less optimal behaviors (e.g., jittering or waving) will be assigned with lower scores (marked with red bounding boxes).

for high-quality demonstrations. Please refer to Sec. D in the Appendix for more training details. The effectiveness of our method is evaluated based on the improvement in success rate after the integration of these newly identified demonstrations. Our experimental results, depicted in Table 4, suggest that L2D substantially improves the performance of imitation learning policy in realistic scenarios. In Figure 5 we visualize the quality prediction for two stacking demonstrations and show that L2D can correctly identify high-quality demonstrations based on sparse labels.

| Environment | Task | Method | High-Quality Demo Selection (%) | SR (%) |
|---|---|---|---|---|
| Simulated | Square | Naive Sampling | 37 | 10 |
| Simulated | Square | Our Method | **95** | **32** |
| Real | Stack | Naive Sampling | 33 | 50 |
| Real | Stack | Our Method | **97** | **80** |
| Real | Lift | Naive Sampling | 33 | 30 |
| Real | Lift | Our Method | **45** | **53** |

Table 4: Performance comparison of naive sampling and our method in terms of high-quality demonstration selection from an unknown dataset and SR increase for two tasks (Can Pick and Place and Lift) on a real robot and one task (Square) in a simulation.

## 6 Limitations

Our method filters out entire demonstrations based on their assessed quality, ensuring that only high-quality data is used for analysis. However, this approach may be overly simplistic, assuming that a low-quality demonstration contains no high-quality segments. In reality, demonstrations often encompass diverse tasks and behaviors, and even a low-quality demonstration may contain sections that could provide useful training data. Additionally, our method assumes a clear distinction between high and low-quality demonstrations, but the line separating the two is often blurred, depending on the complexity of the task and the specificity of the demonstrator's style. Addressing these limitations could lead to even better-learned representations.

## 7 Conclusion

We presented Learning to Discern (L2D), a novel method for efficiently estimating expertise from heterogeneous demonstration data in IL without access to the environment and reward signals. L2D employs sequence-level latent features and temporal embeddings to learn a robust latent representation of complex tasks. With a quality evaluator trained in this latent space, our method can generalize across varied task completion styles and identify high-quality demonstrations, even from new, unseen modes of expertise. This study demonstrates the potential of L2D to filter high-quality demonstrations from vast and diverse datasets, marking a significant advancement in the field of data-centric robotics.

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

# A   Additional Ablation Studies

To further investigate the effect of different architectural components and design choices in our proposed method, L2D, we focus specifically on the roles of S2 contrastive learning and data augmentation. The experimental setup is the same as described in Section 5: identification of high-quality demonstrations from unseen demonstrator experiment on the Square task. We use LSTM+Self Attention architecture with CLS token for contrastive learning followed by a simple MLP encoder for preference learning. The baseline for this experiment does not use S2 contrastive learning or data augmentation. We study the following design choices of L2D: (1) S2.Initial: Initial trajectory segment similarity; (2) S2.Final: Final trajectory segment similarity; and (3) Data augmentations: location-sensitive time warping for segments used in S2 contrastive. As done in Section 5.1, we systematically examine the increase in separability of demonstration quality from new demonstrators by measuring the Wasserstein distances between trajectories of different qualities under different configurations. Higher distance indicates better separability and quality of learned representations.

| S2.Initial | S2.Final | Data Aug | G vs O | G vs B | O vs B | Sum(↑) |
|------------|----------|----------|--------|--------|--------|--------|
| No | No | No | 0.12 | 0.24 | 0.36 | 0.72 |
| Yes | No | No | 0.19 | 0.28 | 0.4 | 0.87 |
| Yes | No | Yes | 0.16 | 0.4 | 0.36 | 0.92 |
| Yes | Yes | Yes | 0.28 | 0.48 | 0.48 | **1.24** |

Table 5: **Impact of Trajectory Segment Similarity and Data Augmentation**. The ablation study demonstrates the value added by incorporating initial and final trajectory segment similarity using S2 contrastive and location-sensitive time warping into our method. Each component enhances the separability of demonstration quality from new demonstrators, leading to superior representations and increased total Wasserstein distance.

Table 5 shows that as we integrate each element of L2D, the total Wasserstein distance increases, indicating improved separability of demonstration quality. The full configuration with both forms of trajectory similarity and time-warping augmentation achieves the highest total distance. This validates the effectiveness of L2D's design choices in improving the quality of learned representations.

# B   Real Robot Demonstrations

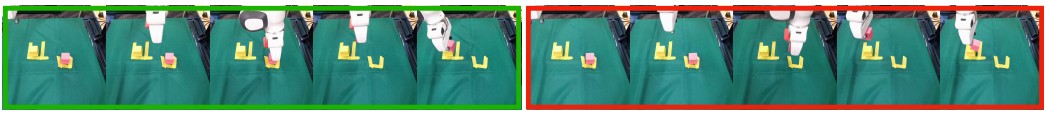

Figure 6: **Real Robot Execution.** We show the rollout results of the policies trained with the demonstrations selected by our method and naive sampling for the Stack task.

We collected physical robot demonstrations for the Lift and Stack tasks and simulated demonstrations for the Square task to provide a more realistic setting for evaluating our method. The Lift task had 60 demonstrations in the training set $D_{known}$ and 60 demonstrations in the test set $D_{unknown}$. The Square and Stack tasks had 100 demonstrations each in $D_{known}$ and $D_{unknown}$. The labeled demonstration data had three quality types: good, okay, and bad. The real robot demonstration data was high-dimensional, containing camera images, gripper position, end-effector pose, joint positions, and joint velocities of the robot arm. After selecting high-quality demonstrations using our L2D approach and baseline approaches, we trained an imitation learning (IL) policy on the best demonstrations from $D_{known}$ combined with the selected demonstrations from $D_{unknown}$. We report the success rate of the trained IL policy on the real robot task.

For the Stack task, we compare the performance of our method with naive sampling. Our results show that policies learned from higher-quality demonstrations exhibit robust and efficient task completion, achieving an 80% success rate over 10 trials. Conversely, policies learned from demon-

strations selected through naive sampling occasionally display peculiar behaviors, such as waving around the target position or dropping the grasped cube at unsafe heights. Consequently, these behaviors contribute to a lower success rate of 50%. Our results demonstrate that the quality of demonstrations plays a crucial role in the performance of learned policies on real robots. By leveraging our L2D approach to identify high-quality demonstrations, we can train policies that achieve more robust and efficient task completion.

## C   L2D Hyperparameters

| Hyperparameter | Default |
|---|---|
| discriminator_learning_rate | 0.0001 |
| label_noise_for_Q | 0.1 [31] |
| discriminator_num_segments | 20000 |
| discriminator_segment_length | 48 |
| discriminator_batch_size | 128 |
| discriminator_training_steps | 500000 |
| initial_trajectory_segment_len | 12 |
| final_trajectory_segment_len | 6 |
| use_pos_encoding | True |
| discriminator_embedding_size | 12 |
| Architecture | [LSTM,2 MLPs,Flatten(), 2 MLPs] [2MLPs] |

Table 6: Hyperparameters used for training L2D to perform Robomimic based ablations.

## D   Imitation Learning Policy Training Details

We choose the default BC-RNN model setting from Robomimic benchmark [11] for learning policies. For simulation tasks the agents only use low-dimensional observation (e.g., robot proprioception, object poses), the epochs are set to 2000 with batch size 100, and we test the agents by running rollout with a maximal horizon of 500 in every 50 epochs. For real-world tasks, we also include the RGB color observations from the three cameras (two at the side, one at the robot's wrist) and the epochs are set to 500. All networks are trained using Adam optimizers [41].

| Hyperparameter | Default |
|---|---|
| Batch Size | 16 |
| Learning rate (LR) | 1e-4 |
| Num Epoch | 1000 |
| Train Seq Length | 10 |
| MLP Dims | [400, 400] |
| Image Encoder - Wrist View | ResNet-18 |
| Image Feature Dim | 64 |
| GMM Num Modes | 5 |
| GMM Min Std | 0.01 |
| GMM Std Activation | Softplus |

Table 7: Hyperparameters for L2D

For the Robomimic Square task on Multi-Human Dataset experiments in Section 5, there is an equal 150-150 demonstration split. L2D trains on 150 known demonstrations and is tasked to select the demonstrations that belong to the "good" quality from the set of 150 unknown demonstrations. Familiar demonstrator experiment in Table 2, aims to test the IL policy's performance when it has access to a combination of known and unknown demonstrations. Specifically, BC-RNN is provided with 100 demonstrations: the top 50 from $D_{known}$ and another 50 chosen from $D_{unknown}$. For the unfamiliar demonstrator experiment in Table 3, BC-RNN trains exclusively on 50 demonstrations selected from $D_{unknown}$.

Once we have filtered demonstrations using different methods (including proposed method and baselines) and consistent heuristics across methods, we employ Behaviour Cloning for training. We use RNN-based behavioral cloning (BC-RNN) to evaluate algorithm performance using subsets based on demonstrator quality as prior works show [6, 11, 16] it is effective at handling mixed quality data.

