# OpenReview forum: "Learning to Discern: Imitating Heterogeneous Human Demonstrations with Preference and Representation Learning"
_robot-learning.org/CoRL/2023/Conference — CoRL 2023 Poster_

### Official Review · Reviewer_xRQR · 2023-07-17

**Confidence:** 4
**Originality:** Good
**Technical Quality:** Fair
**Clarity Of Presentation:** Fair
**Impact:** 3

**Recommendation:**

Weak Accept: I recommend accepting the paper, but will not argue for my recommendation if the majority of other reviewers have a different opinion.

**Review:**

I like the overall idea; however, there are lots of typos and the paper is lacking many important details. It is unclear whether Strategy 1 and 2 are used together or whether one is chosen. The domains deserve more description before discussing their nuances. Also, the way in which the latent space and quality predictor are trained and used is not rigorous enough and makes it hard to understand. I like that the authors use real demonstrations and test on a real robot. The problem is important and the method seems to work well, but the writing needs work and many details are missing.

---
Post rebuttal: I still think there are many details lacking: data on users who provided demonstrations, specific formulas used for the different training losses, how the Wasserstein distance was calculated, pseudo-code for the different parts of the algorithm, etc. I am updating my score to a weak accept, since I think these issues can be corrected in the camera ready draft. I strongly encourage the authors to significantly expand the appendix to foster reproducibility as well as to release code since it seems very hard to reproduce given how the paper is currently written.

**Quality Of The Limitations Section:**

Additional details required

**Questions For Rebuttal:**

Given that you have quality labels already. Why not train a classifier to map directly from segments to qualities and then just train on new demos that have high-quality predicted labels? This seems like it could be a strong baseline to compare against. Also, why not just compare against something trained to map directly from segments to scores via preference learning and use the raw scores without the GMM to compute better and worse trajectories?

Fig 3. It is not clear what G vs O or G vs B represent. Also the captions say Good, Okay, Poor, but not Bad.

No discussion of user study details, IRB approval, participant information, number of participants, training routine, etc.

It is unclear what the ablations mean and how they are implemented. Also, I see that they lead to better distance in the latent space, but I don't see where it is shown whether this distance correlated with higher success.

It is unclear what the tasks are, but the authors assume familiarity. Please add figures and descriptions before discussing nuances of the environments. E.g., line 108 talks about square but this doesn't make sense unless readers are already familiar with this env, which they probably aren't. Also, bottleneck states are very important and I don't see why they are indifferent to demonstration quality. Usually bottle neck states are the ones that actually matter.


Line 208. Need to reference authors or algo name for [6] and describe this baseline. Where are the results for this claim?

Line 209 What is SR?

What kind of labels are provided? Good, Okay, Bad? Is this provided for each trajectory or each segment? Based on section 3 is seems that each trajectory gets a high-level quality label, but then how do you get D_A, D_B, etc which seem to correspond to trajectory segments of the same quality.

Line 110: How do you know how to sample quality influencing trajectory segments? Do you train a classifier?

Strategy 2 is confusing as written. Why not sample from final segment or same region?

Limitations talks about using high-quality sections of bad demos. This seems possible since the training data has segments of good and bad portions of full trajectories and that the method predicts the quality of segments so why not do this? What is the barrier?

How is the GMM used to predict a trajectory-level preference label? It seems that it is trained on trajectory snippets.

How is the encoder and preference model trained? What are the loss functions?

What is the heuristic used for determining whether the demonstration should be used for training the policy?

**Robotics Focus:**

Sufficient demonstration on hardware

**Summary Of Paper:**

The authors propose a method to learn from a variety of demonstrations. The approach uses a small number of quality labels on demonstrations to train a latent space and an evaluator that uses sequence-level latent features to predict demonstration quality. By training only on good demonstrations, the authors show improvements over prior work.

**Summary Of Recommendation:**

I think the paper addresses an important problem. But, as written, I do not think it is ready for publication. There are too many missing details that make it hard to evaluate.

--
post rebuttal: the paper has improved but can still be further improved with more details. however, the overall idea has merit so I am leaning towards acceptance, but wouldn't argue against rejecting the paper.

---

### Official Review · Reviewer_pGpd · 2023-07-21

**Confidence:** 4
**Originality:** Good
**Technical Quality:** Good
**Clarity Of Presentation:** Fair
**Impact:** 3

**Recommendation:**

Weak Accept: I recommend accepting the paper, but will not argue for my recommendation if the majority of other reviewers have a different opinion.

**Review:**

### Quality and Clarity

Overall, the writing in the paper needs a lot of improvement to make it clear. Currently, the methods and evaluations are not as clear as they should be, and this is impacting the quality of the paper as well. There could be an opportunity to tighten this up during the rebuttal phase. The details are listed below:

Line 138 to 143 (Training Quality Critic): It’s unclear if this is a classifier or a regressor (since many other parts of the text talk about scalar values being assigned to the quality labels). Similarly, is the added label noise in the discrete label space or the continuous scalar space?

Line 176: “filtered demonstrations”: the filtering procedure is not clear. In the previous section, the paper says that “we use a heuristic to determine whether the demonstration should be used for training the policy”, and provides an example heuristic. However, for experiments, a description of the actual heuristic used needs to be shown so that the results can be correctly interpreted.

Line 177: “different methods” – it’s unclear what methods we’re talking about here. Maybe you could instead say that the “filtered demonstrations” are obtained through different methods, including the baselines and the proposed method.

Line 187: “quality subset distributions” – it’s unclear what this means. It’s probably better described as  “histogram of estimated values Q() for each subset of trajectories with specific labels”.

Section 5.1: The baselines should be described better. What is “the basic preference learning approach” (line 190)? Does it involve passing the trajectory directly into the quality critic, instead of the embedding? Or is it a prior method from the literature?

Figure 3: This should probably be split up into a separate table and figure. The table should be captioned separately. “G vs O” etc titles are unclear; you should be able to fit in “good vs ok” and have a better description in the caption as to what the numbers mean.

Figure 3: These distributions seem to show a bucketed version of an estimated scalar value (since we see multiple bars for each integer). However, the previous section (line 156, “Estimating Quality of Unseen Demonstrations”) says that each demonstration is evaluated by sampling segments from it and getting a scalar score from each of them. How is an aggregate value obtained for a single demonstration? Or does the histogram show the estimated values of each segment? It would also be good to understand the actual labels that were used (i.e., was it bad = 1, ok = 2, good = 3?)

Main results: (paragraph starting line 201) – the description of the experiment is not very clear. It should probably be mentioned that $D_{known}$ is used for training, and that the “pool” being tested is $D_{unknown}$, but that’s just my assumption – it should be clarified in the text.

Line 208: Please use the term ILEED instead of just “[6]”, since you also use it in the table and elsewhere.

Line 209: “SR” is used before defining it. Also, this section should refer to Figure 4 in the text, since that’s the raw data based on which the claim is made.

Figure 4: The table on the right needs to be explained in more detail – it looks like you run each method on the entire dataset (training and test), and look at the ground truth labels of the top 100/300 high-quality trajectories? (I tried to infer this from the values in the table, but please describe this explicitly). Or is it the top 100/150 from $D_{unknown}$? (In any case, it should probably be restricted to $D_{unknown}$ since evaluating on the training set would not be informative).

It would also be good to evaluate this on more than just a single task from Robomimic – at least expanding this evaluation to the other three Robomimic tasks would convince the reader that this result and algorithm can work in various settings.

Ideally, it would be great to also include a result from behavior cloning on *all* demonstrations as a baseline, not just the naive sampling. Yes, it will be trained on a larger number of demonstrations than the others, but in my opinion this would be a fairer (and more realistic) comparison than randomly sampling a smaller set, since the full dataset is available and there’s no reason to subsample if quality predictions are not available. This should be the bar to beat – by sub-selecting the “good data”, can you outperform a model that’s simply trained on all the data?

Sec 5.3: Learning from real-world demos: Please specify the number of demonstrations per task that were used for training and test.

Line 239: please add the table number as reference.

There are many inconsistencies in the specification of which tasks were actually conducted in the real world. Line 231 says “tasks such as Square Nut Assembly, Lifting, and Stacking in both simulated and real-world environments”. The Table 1 Caption says that only Lifting was in the real world, and the other were in sim. And the table itself says that Lift and Stack were in the real world, while Square was in sim. The level of inconsistency here, combined with the lack of any real-world videos leaves me suspicious about which experiments were actually conducted in the real world. If the first two rows of Table 1 were indeed in a simulated environment, what is this result trying to show over what was already shown in Section 5.2? Is this about using an image-based end-to-end policy in simulation, vs the previous section being state-based?

Also, line 236 (in the real-world experiment section 5.3) is where I realized for the first time that the previous experiments (5.1, 5.2) were not using images. The inputs/outputs/action spaces for the tasks are not described anywhere, but they need to be.

Many tables are captioned as “Figure xx” - please fix them, and the references to them (some references are called Table xx, which currently don’t exist.)

### Originality

The proposed approach has novelty compared to prior approaches, at least based on my knowledge of the field. However the quality and clarity issues mentioned above significantly weaken the paper.

**Quality Of The Limitations Section:**

Limitations are addressed clearly

**Questions For Rebuttal:**

The detailed list of quality/clarity issues, and some experimental issues were listed above, and could be fixed during the revision period.

**Robotics Focus:**

Highly relevant to robotics but no hardware experiments

**Summary Of Paper:**

Imitation learning requires large diverse datasets, but these datasets will come from a variety of demonstrators, and the quality cannot be guaranteed to be uniformly high. This paper proposes a method to select high-quality demonstrations that can be used for training policies. The approach relies on a small subset of the demonstrations being labeled with their quality, from which a latent representation is learned, followed by a quality critic. The critic is then used to pick the top quality trajectories from the unlabeled set, which can then be used to train a policy.

Simulation results show that the proposed method outperforms previous work as well as different ablations of it. Some real-world experiments exist as well, but the details are unclear (see details below).

**Summary Of Recommendation:**

The paper has good ideas, but is significantly weakened by its presentation. Many critical aspects of the algorithmic and experimental details are not provided in the main text or the appendix. There are also severe inconsistencies about which experiments were actually conducted in the real-world, which leads me to discount those results entirely until more evidence is shown.

---

### Official Review · Reviewer_kfod · 2023-07-29

**Confidence:** 3
**Originality:** Good
**Technical Quality:** Good
**Clarity Of Presentation:** Good
**Impact:** 2

**Recommendation:**

Weak Accept: I recommend accepting the paper, but will not argue for my recommendation if the majority of other reviewers have a different opinion.

**Review:**

I think the paper was well organized and fairly easy to read and well motivated. While the individual concepts introduced by the paper are not particularly novel, the motivation behind putting them together (for the most part) to solve the problem of imitation learning from heterogenous distributions is explained convincingly. The experiments and the ablation studies  were also very interesting. However, there are some foundational steps (see questions for rebuttal) that are unclear.

**Strengths:**
- ablation studies and experiments look very convincing
- results on hardware

**Weaknesses:**
- Some clarity issues (See questions for rebuttal).
- I am not very convinced with the choice of baselines. ELICIT does not seem very suitable as a baseline since the objective of the method is quite different, maybe something like CAIL and TREX is better suited.

**Quality Of The Limitations Section:**

Limitations are addressed clearly

**Questions For Rebuttal:**

    1. Why is it important to separate the demonstrated trajectories into subsets D_A, D_B, D_C etc.? If there are a limited number of individual labels, why can these be not directly used to train the quality critic? Related questions:
        1. How many individual labels and how many subsets are there in your experiments
        2. How are the subsets chosen or arranged?
    2. From the section “Training Quality Critic”, the quality critic seems to be a multi-class classifier to me. I do not understand then why comparing pairs of datasets necessary (lines 142-143) . Also confusing - Is the input to the quality critic the trajectory segment or a pair of trajectory segments?
    3. CAIL sounds like a better baseline for comparison to me – since it also assumes access to demonstration labels. The authors have mentioned this paper in related works and seem to have dismissed it as a baseline citing its requirement for “dense labels”. However, the authors also assume access to quality labels for each of the demonstrations in their known dataset.
    4. Are the suboptimal demonstrations simply used for training the quality critic and not in any way used for imitation learning?

**Robotics Focus:**

Sufficient demonstration on hardware

**Summary Of Paper:**

This paper learns to identify  the quality of an unseen demonstration and thereby decide whether to use it for imitation learning. To do so, this work learns a quality critic from a small set of ranked demonstrations. This quality critic can evaluate segments of length L from each demonstration, which are encoded into a position-aware latent representation. The latent representation is trained first using contrastive strategies.  The quality critic is trained by casting the predicted labels from all the segments in the demonstration into a Gaussian Mixture Model  where the average predicted label from the GMM is matched with the true demonstration label. During test time, the suitability of an unseen demonstration for imitation learning is decided by counting the quality critic labels assigned to the highest quality set.

**Summary Of Recommendation:**

The paper was easy o read and well motivated in my opinion with nice experiments, but still has some clarity issues that need to be addressed.

---

### Official Review · Reviewer_L98X · 2023-07-30

**Confidence:** 3
**Originality:** Fair
**Technical Quality:** Fair
**Clarity Of Presentation:** Fair
**Impact:** 2

**Recommendation:**

Weak Reject: I recommend rejecting the paper, but will not argue for my recommendation if the majority of other reviewers have a different opinion.

**Review:**

The clarity of the paper can be significantly improved. At the end of the review is a list of specific suggestions and recommendations

Utilizing a contrastive loss to embed demonstration segments into a latent space proves to be a valid approach. Prior works, such as [1], have explored the use of embedding task progression for agents with varying embodiments. Overall, this topic holds significant relevance, and further research in this direction could offer valuable insights for the field of imitation learning.

In Lines 128-129, the authors assert that preference learning is not effective in manipulation tasks. However, this claim appears to be at odds with the conclusions drawn in various papers on preference learning [2][3][4], where manipulation tasks have been explicitly taken into account. Therefore, it is essential for the authors to acknowledge and discuss these relevant works in the Related Work section

In the paper, the authors made reference to a relevant work [5], which could have served as a suitable baseline for their work. Unfortunately, they did not include a comparison with this paper. Additionally, the lack of clarity in their writing makes it difficult to provide further comments on their results. Specific points can be found at the end of this review.

The authors should provide reasoning for choosing BC-RNN as their BC method. Moreover, an ablation study encompassing various BC approaches would constitute a valuable contribution.

Presently, the authors assess their method under the condition where an equal number of demonstrations exist within each quality group, namely "good," "okay," and "bad." However, it is beneficial to extend the evaluation to encompass scenarios where the relative quantity of "good" demonstrations is reduced. Such case is relevant as it mirrors real-world situations in numerous tasks.

Suggestions for improving the clarity:

In Line 18, the authors introduced the acronym IL for imitation learning; however, they do not use it consistently afterwards. Overall, the authors are advised to adhere to the standards of academic writing when using acronyms

Figure 3 consists of a table and a figure, and should be appropriately separated. In Line 189, authors referenced Table 3, but in the paper no table is labeled in such manner. Most likely the authors discussed the table in Figure 3, but that should be clearly written.  Moreover, Figures 4 and 5 are actually Tables. Most of the figures are never referenced in the paper.

Authors introduce some notation which can be confusing when considering other papers in the domain. Specifically, they represent the preference network with 'Q' (line 95), which is commonly used to denote the state-action value function in Reinforcement Learning

Minor writing comments:

Line 108: the authors mention the task Square, a reference and short short description of the task would increase the readability.

Line 172: A reference to the robomimic dataset should be included.

Line 248: Sentence does not have a verb in it.

Line 294: Name of an author is unnecessary bold.

[1] K.Zakka, A. Zeng, P. Florence, J. Tompson, J. Bohg, D. Debidatta, XIRL: Cross-embodiment Inverse Reinforcement Learning, CoRL 2021

[2]K. Lee, L. Smith, P. Abbeel, PEBBLE: Feedback-Efficient Interactive Reinforcement Learning via Relabeling Experience and Unsupervised Pre-training, ICML 2021

[3]A. Taranovic, A. Kupcsik, N. Freymuth, G. Neumann, Adversarial Imitation Learning with Preferences, ICLR 2023

[4]C. Kim, J. Park, J. Shin, H. Lee, P. Abbeel, K. Lee, Preference Transformer: Modeling Human Preferences using Transformers for RL, ICLR 2023

[5] D. S. Brown, W. Goo, and S. Niekum. Better-than-demonstrator imitation learning via automatically-ranked demonstrations, ICLR 2019

**Quality Of The Limitations Section:**

Additional details required

**Questions For Rebuttal:**

Could the authors reference results presented in the tables in the text of the paper?

Could the authors evaluate the additional baseline that is mentioned in the review?

Why did the authors choose BC-RNN algorithm? Did they consider other BC methods?

Please improve the writing by addressing points in the review.

**Robotics Focus:**

Sufficient demonstration on hardware

**Summary Of Paper:**

In their paper, the authors propose a novel method for learning a latent representation of demonstrations, which temporally embeds their segments. The learned latent space is then used to train a preference model, enabling the discernment of high-quality demonstrations. Subsequently, the authors employ these filtered demonstrations to train an agent using a behavior cloning method. To evaluate their proposed approach, the authors conducted experiments in simulation as well as on a real robot, demonstrating its superior performance when compared to the baseline methods

**Summary Of Recommendation:**

The paper is not clearly written, evaluation is incomplete, and the contribution is not significant.

---

### Author Response · Authors · 2023-08-12
**Common Response to Reviewers**

We thank the reviewers for their insightful comments and valuable feedback. We have uploaded the updated version of paper using the rebuttal button for every reviewer. First a common response, then per-reviewer responses. Following clarifications are inline with updated version of the paper:

**Comparison with TREX[1] as a baseline**

[LINE 216] Preference Learning baseline is a best-effort adaptation of TREX [1].

TREX begins by learning a reward model using human preferences, a procedure we evaluate in our own setup. Following this, TREX carries out Inverse Reinforcement Learning (IRL) to cultivate a policy. On the contrary, L2D is not rooted in IRL. It's an offline data filtering method. A critical distinction between the two is that, unlike TREX, L2D cannot access the environment to gain feedback. Therefore, a direct comparison with TREX isn't feasible; we can only draw parallels with its reward-learning neural network. Our Preference Learning baseline is this best-effort adaptation of TREX

**Real Robot Experiment Video**

Please find the video for Stacking Task on real robot on L2D website: https://sites.google.com/view/learning2discern

**Dividing known dataset D_known into sub-datasets {D_A, D_B, D_C, …}**

[Line 95-98] Every trajectory is accompanied by a label $l \in L$. D_known is divided into sub-datasets of similar quality levels {D_A = “good”, D_B = “okay”, D_C = “bad”}. The number of such sub-datasets is equal to the cardinality of L. Each sub-dataset contains trajectories corresponding to a unique given label such that all trajectories with that label are assigned to that sub-dataset exclusively.

The strength of L2D is that our method does NOT require dense rankings among demonstrations (i.e., 1 >2> 3 > .... >N ) . Our method requires “coarse rankings”, i.e., demonstrations to be placed in quality groups {D_A, D_B, D_C, …}.

**Training and inference of Quality Critic Q**

[LINE 158-164] $Q: R^d \rightarrow {R}\$, is a regressor, mapping the d-dimensional trajectory segment embedding to a continuous scalar value indicative of the quality of the trajectory segment. It uses the pairwise ranking loss [1, 2]. In the training phase, we sample pairs of trajectory segments, $\(\sigma_A\)$ and $\(\sigma_B\)$, randomly sampled from sub-datasets $\(D_A\)$ and $\(D_B\)$, respectively. During the inference stage, $Q()$ estimates the quality score of individual segments.

**Robomimic Task and Dataset**

[Line79, Fig3] We have introduced a paragraph on Robomimic [3]  in the related work section, detailing its corresponding tasks and datasets. Within the experimental setup, we've included a diagram illustrating the demonstrations for the square task to enhance the reader's understanding of the demonstration quality.

**Robomimic Task Sub-dataset setup**

[Line 82] All Robomimic task setups feature three quality groups: good, okay, and bad. Each group contains 50 demonstrations for both $D_{known}$ and $D_{unknown}$. These group categorizations and demonstrations adhere to the specifications of the Robomimic dataset [3]

References:

[1] TREX Extrapolating Beyond Suboptimal Demonstrations via Inverse Reinforcement Learning from Observations ICML

[2] Deep reinforcement learning from human preferences NeurIPS

[3] Mandlekar A, et al. What matters in learning from offline human demonstrations for robot manipulation. CoRL 2021

---

### Decision · Program_Chairs · 2023-08-30

**Decision:**

Accept (Poster)

**Comment:**

The paper offers an intersting approach to offline imitation learning with convincing experiments. While initially, the reviewers have identified several concerns regarding the choice of the baseline, the number of experiments (e.g. for robomimic) and the presentation of the approach, these concerns were adequately addressed by the reviewers. The authors clarified their choice of baselines which and added new experiments with additional imitation learning datasets. While there is still one reviewer with a "weak reject", I consider most of the points addressed, where the only comment not addressed is the use of more recent imitation learning algorithms as backbone (e.g. diffussion policies have shown to outperform BC-RNN by a large margin). Yet, this can be left as future work in my opinion and I would still recommend the paper for publication.